# Granular Computing-driven SAM: From Coarse-to-Fine Guidance for Prompt-Free Segmentation

## Abstract

Prompt-free image segmentation aims to generate accurate masks without manual guidance. Typical pre-trained models, notably Segmentation Anything Model (SAM), generate prompts directly at a single granularity level. However, this approach has two limitations: (1) **Localizability**, lacking mechanisms for autonomous region localization; (2) **Scalability**, limited fine-grained modeling at high resolution. To address these challenges, we introduce Granular Computing-driven SAM (**Grc-SAM**), a **coarse-to-fine** framework motivated by **Gr**anular Computing (GrC). First, **the coarse stage** adaptively extracts high-response regions from features to achieve precise foreground localization and reduce reliance on external prompts. Second, **the fine stage** applies finer patch partitioning with sparse local swin-style attention to enhance detail modeling and enable high-resolution segmentation. Third, refined masks are encoded as latent prompt embeddings for the SAM decoder, replacing handcrafted prompts with an automated reasoning process. By integrating multi-granularity attention, Grc-SAM bridges granular computing with vision transformers. Extensive experimental results demonstrate Grc-SAM outperforms baseline methods in both accuracy and scalability. It offers a unique granular computational perspective for prompt-free segmentation.

## 1 Introduction

Semantic segmentation, as a core task in computer vision, aims to assign semantic category labels to each pixel in an image Geng et al. (2018). In recent years, the rise of deep learning has significantly advanced this field. Particularly, the introduction of Transformer-based models to segmentation has enhanced long-range dependency modeling capabilities through their self-attention mechanisms, while also improving robustness and generalization performance Lateef & Ruichek (2019). Despite these advances, existing segmentation models still require retraining for specific tasks, lacking unified generalization capabilities and cross-domain adaptability.

With the emergence of vision foundation models, the paradigm of segmentation has begun to shift. Meta AI's Segment Anything Model (SAM) Kirillov et al. (2023) is the first general-purpose promptable segmentation model. By leveraging large-scale data and powerful Transformer architectures, SAM demonstrates strong transferability in open-world scenarios. It plays a vital role in applications such as image understanding Kweon & Yoon (2024), autonomous driving Yan et al. (2024), medical imaging Gao et al. (2024), and remote sensing Zhang et al. (2024). Its core idea is to guide segmentation through diverse prompts (points, boxes, masks), thus reducing reliance on task-specific supervision. This paradigm of promptable segmentation not only strengthens the generalization of segmentation methods but also broadens their applicability in domains such as medical imaging, remote sensing, and video understanding.

Nevertheless, recent surveys highlight that SAM still struggles with fine-grained structures and semantically complex scenes Zhang et al. (2023b). Its results often lack precision in boundary delineation and small-object recognition, suggesting that bridging general-purpose segmentation with the fine-grained requirements of semantic tasks remains an open challenge Zhang et al. (2023b). First, segmentation tasks require fine-grained spatial representation, such as distinguishing object bound-

aries, adjacent small objects, and complex textures. Existing models often emphasize global semantics but fail to preserve local details. Second, while SAM exhibits strong generalization in large-scale open scenarios, its mask generation mechanism heavily relies on global attention and dense prediction. This design frequently leads to boundary smoothing, missed details, and poor recognition of small targets. Moreover, derivative works (FastSAM Zhao et al. (2023), MobileSAM Zhang et al. (2023a), HQ-SAM Ke et al. (2023)) mainly focus on speed or resolution improvements, without deeper exploration of hierarchical region modeling. In other words, future frameworks must integrate multi-granularity approaches, combining coarse-grained localization with fine-grained reasoning to achieve high-quality analysis in complex scenarios while maintaining efficiency.

A deep analysis of SAM's design reveals two major structural limitations. First, SAM relies on manually provided prompts, limiting its application in fully automated scenarios. In many real-world contexts, such human interaction is impractical, while pixel-level annotation incurs prohibitively high costs. Second, SAM's global self-attention mechanism suffers from inefficiency on high-resolution images. While global attention helps maintain semantic consistency, aggressive downsampling inevitably leads to detail loss. With the growing exploration of the SAM, numerous studies have focused on reducing reliance on manual prompts to enhance practicality and automation. For instance, AoP-SAM Chen et al. (2025) automates prompt generation, Talk2SAM incorporates text-guided semantics, HSP-SAM Zhang et al. (2025b) introduces hierarchical self-prompting, MaskSAM Xie et al. (2024) models prompts as mask classification, and SAM-CP Chen et al. (2024) leverages composable prompts for more flexible segmentation. Further works such as BiPrompt-SAM Xu et al. (2025), EviPrompt Xu et al. (2023), IPSeg Tang et al. (2025), Self-Prompt SAM Xie et al. (2025), and Part-aware Prompted SAM Zhao & Shen (2025) explore diverse strategies for automatic or adaptive prompting. These studies collectively reveal a clear trend: toward prompt-free or minimally interactive segmentation, which is particularly crucial in scenarios where manual prompts are difficult to obtain. Against this backdrop, our work aims to further advance prompt-free segmentation while integrating granularity-controllable and semantic-enhancement mechanisms to achieve more efficient and generalizable segmentation.

In addition, the performance of SAM largely depends on the type and coverage of input prompts Yuan et al. (2024), Cheng et al. (2023). Existing research indicates that in most scenarios, bounding box prompts typically yield higher segmentation accuracy than single-point prompts, while point prompts only approach the accuracy of bounding box prompts when their quantity is significantly increased Chen et al. (2025). While box prompts and point prompts can be combined to improve accuracy, they cannot be applied simultaneously Mazurowski et al. (2023). In contrast, dense spatial priors and boundary constraints provide the decoder with stronger semantic and geometric information, enabling the generation of finer-grained mask results Jiang (2025). This advantage led us to directly generate the mask prompt within GrC-SAM rather than deriving it from point prompts, thereby enhancing localization accuracy and boundary clarity.

SAM also provides an official automatic mask generation mode (AMG), which removes manual interaction by internally sampling dense point prompts to produce candidate masks. However, AMG remains a heuristic post-processing procedure and exhibits three fundamental **limitations**: **(1)** uniform point sampling does not prioritize semantically salient regions, leading to inaccurate coarse localization; **(2)** generating and filtering hundreds of candidate masks incurs substantial computational and post-processing overhead; and **(3)** the process still struggles to capture high-resolution details, often resulting in blurred boundaries. These limitations motivate the need for an internal, learnable mechanism that can provide semantically guided region selection without relying on externally simulated prompts.

**Motivation:** To address the aforementioned limitations, this study proposes a prompt generation mechanism based on granularity computation, enhancing both model performance and automation levels. Inspired by granular computing Fang et al. (2020) and prompting-driven Fang et al. (2025), we adopt a conceptual coarse-to-fine framework: the coarse processing stage rapidly locates potential target regions, while the fine stage models details through a local attention mechanism. Crucially, this framework does not directly output segmentation results but focuses on generating high-quality mask prompts and providing efficient region guidance. It ingeniously integrates granular computation with characteristics of human visual cognition into segmentation tasks. It rapidly directs attention to critical regions requiring fine-grained processing while filtering out vast amounts of irrelevant information in complex large-scale scenes. By dynamically allocating computational re-

sources to key areas, it reduces overall computational cost while enhancing fine-grained processing capabilities in critical regions while maintaining global perception.

Based on the above motivations, this study advances automation, efficiency, and accuracy in general segmentation through four major **contributions**. **First**, a **granular computing-driven automatic prompt generation framework** is proposed. This design guides segmentation tasks without human intervention, enhancing automation while achieving more precise downstream segmentation. **Second**, the **global semantic information extraction mechanism** enhances the representation of boundaries and fine details while ensuring semantic consistency. **Third**, a **sparse attention variant** is introduced. This approach reduces computational cost while maintaining semantic awareness, enabling efficient and accurate processing of high-resolution and detail-rich regions. **Finally**, a **granularity-computation theoretical foundation** is established for automatic segmentation in complex scenarios, demonstrating how the proposed granular computing-driven framework extends the applicability of SAM and its derivatives to cross-domain, fine-grained, and efficient segmentation tasks.

## 2    RELATED WORK

**SAM and its granularity derivative models:** SAM includes an official automatic mask generation mode (AMG), which simulates manual prompting by uniformly sampling dense point prompts and generating a large set of candidate masks. These candidates are ranked, refined, and filtered using non-maximum suppression. Although AMG removes the need for manual prompts, it relies on brute-force exploration of point prompts and thus suffers from three limitations: (1) uniform sampling fails to prioritize semantically salient regions, (2) evaluating hundreds of candidate masks introduces significant computational overhead, and (3) fine-grained boundaries are not well preserved at high resolution.

Recent advances in segmentation models based on SAM have significantly improved performance. At the same time, research efforts are increasingly focusing on granularity-based approaches. Fast-SAM Zhao et al. (2023) accelerates inference through lightweight design; MobileSAM Zhang et al. (2023a) optimizes for resource-constrained devices; HQ-SAM Ke et al. (2023), SAM-Adapter Chen et al. (2023b), and SEEM Zou et al. (2023) enhance mask resolution or segmentation accuracy; Med-SAM Ma et al. (2024) and SegGPT Wang et al. (2023) adapt SAM to medical imaging or multimodal scenarios. Most models still rely on global attention mechanisms and dense predictions, which may cause boundary smoothing and overlook fine structural details. To overcome the limitations of global attention and dense predictions, recent studies have explored segmentation frameworks with granularity control and semantic enhancement. GraCo Zhao et al. (2024) proposes an interactive mechanism to control segmentation granularity; Semantic-SAM Li et al. extends SAM toward joint "segmentation + recognition" across arbitrary granularities; Fine-grained All-in-SAM Li et al. (2025) leverages part-level prompts or molecular priors to enhance fine boundary delineation and class discrimination; and SARFormer Zhang et al. (2025a) introduces a semantic-guided Transformer to reinforce cross-granularity context modeling. Collectively, these works demonstrate a clear trend: through granularity control and semantic enhancement, SAM is evolving from "segmenting any object" toward "understanding any scene with multi-granularity and multi-semantics," thereby providing richer structural information for downstream tasks.

**Patch-based Vision Transformers:** Granularity computation emphasizes organizing and processing information through multi-level, multi-granularity approaches Shi & Yao (2025). Coarse-grained representations provide global semantics, while fine-grained representations preserve local details Zhang et al. (2023d). Granularity structures facilitate complementary relationships and transitions between different granularity levels. This concept finds natural expression in the visual domain: patch-based visual transformers partition images into fixed-size patches, with different patch sizes corresponding to distinct granularity levels. FlexiViT Beyer et al. (2023), DG-ViT Song et al. (2021), DW-ViT Ren et al. (2022), and NaViT Dehghani et al. (2023) demonstrate the potential to balance coarse-grained semantics with fine-grained information through dynamic adjustment of patch sizes. Conversely, Medformer Wang et al. (2024), CF-ViT Chen et al. (2023a), and DVT Wang et al. (2021) achieve top-down information guidance via hierarchical interactions of multi-granularity features. Studies such as TCFormer Zeng et al. (2022), SCA Liu et al. (2023), MPA Liu et al. (2016), and PMT Sun et al. (2025) further emphasize the fine-grained modeling

of critical regions or minute objects, often integrating multi-granularity processing or boundary enhancement strategies. From a granularity perspective, these methods can be abstracted as hierarchical coarse-to-fine processing of image information: first locating potential target regions using coarse-grained features, then refining details through fine-grained or local mechanisms.

**Sparse Attention Mechanism from a Granular Computation Perspective:** Traditional self-attention mechanisms incur substantial computational and memory overhead in visual tasks, whereas sparse attention achieves "on-demand modeling" by selectively establishing dependencies. This approach aligns closely with the granular computing philosophy of "coarse-grained yet refined, hierarchically organized" processing. Existing research has proposed multiple sparse models: Sparse Transformer Child et al. (2019) and Longformer Beltagy et al. (2020) combine local windows with skip connections; BigBird Zaheer et al. (2020) balances local, global, and random connections. In the visual domain, methods like Swin Transformer Liu et al. (2021) sliding windows and cross-window mechanisms to explicitly introduce hierarchical local-global modeling. Reexamined through the granularity computation lens, these approaches establish granularity hierarchies between coarse-grained (global dependencies) and fine-grained (local windows), forming cross-level information aggregation structures.

## 3 OUR APPROACH

### 3.1 OVERVIEW

The GrC-SAM method directly embeds a tightly coupled mask generator module into the original SAM architecture rather than treating it as a separate post-processing tool. This module directly utilizes the multi-layer attention features from the image encoder. Inspired by Zhang et al. (2023c), for deep networks composed of stacked multi-head attention modules, attention patterns in shallow layers are often unstable, with performance gains primarily driven by deep attention weights. This module automatically generates latent mask prompts based on attention scores, which are then processed through granularity-based refinement before being fed into the prompt encoder and mask decoder. This achieves prompt generation and segmentation prediction within a unified end-to-end framework, eliminating the loose coupling between "sampling and post-processing" in SAM-AMG while ensuring consistency between training and inference stages.

### 3.2 GRANULAR COMPUTING-DRIVEN COARSE-TO-FINE FRAMEWORK

We formulate a general framework for coarse-to-fine image segmentation under the perspective of granular computing. The key idea is to define hierarchical granularity spaces and mappings that guide the segmentation process from coarse regions to finer details.

**Definition 1** Given an image domain $X$, we define the granularity set $\mathcal{G} = \{G_c, G_f\}$, where $G_c$ denotes the coarse-grained space and $G_f$ denotes the fine-grained space. These granularity spaces correspond to different levels of partitioning the image into patches: e.g., $G_c = \{U_j, patch\_size_{coarse}\}$ and $G_f = \{U_j, patch\_size_{fine}\}$.

We introduce two mappings between these spaces: $\phi : X \to G_c, \psi : G_c \to G_f$, where $\phi$ maps the input image $X$ to the coarse-grained space $G_c$, and $\psi$ fine-grained $G_c$ into the finer granularity space $G_f$. The green section of Fig. 1 roughly illustrates the coarse-to-fine framework. We will describe it in more detail using formal mathematical expressions.

**(1) Coarse-grained space** $G_c$ In the coarse stage, the image is partitioned into large patches that form the coarse granularity space $G_c$. For each patch, we compute a semantic importance score by fusing multi-layer attention responses from the encoder. This fused score highlights regions that consistently receive high attention across deeper layers. A learnable threshold is then applied to obtain a soft, differentiable coarse mask, which down-weights irrelevant or noisy regions while preserving high-response areas. The resulting coarse mask $M_c$ serves as a spatial prior, indicating where fine-grained processing should be allocated. In essence, the coarse stage provides a global, semantically guided localization signal without committing to final segmentation boundaries.

Figure 1: **GrC-SAM Model Architecture Diagram.** We directly embed the granularity computing-driven masking generator into SAM. Specifically, it is positioned between the image encoder and the prompt encoder. Guiding information is extracted from the multi-layer attention scores of the image encoder, enabling the generation of masking prompts through granular computing-driven principles and a local sparse attention mechanism.

**(2) Fine-grained space** $G_f$. The fine stage focuses on regions highlighted by the coarse mask and subdivides them into finer patches to capture detailed structures. Within these selected regions, we apply local windowed attention to model boundary details and fine-scale variations. Each fine patch receives an attention response that reflects its local relevance. A second learnable threshold adaptively filters these responses, producing a soft fine mask that emphasizes truly informative areas while suppressing residual noise from the coarse stage. Conditioned on $M_c$, the fine-grained representation $M_f$ provides refined spatial guidance with enhanced boundary precision and local context modeling.

**(3) Recursive coarse-to-fine relation.** The entire process can be summarized as a hierarchical, recursive mapping: $M_c = f_\theta(\phi(X)), M_f = g_\theta(\psi(G_c) \mid M_c)$.

## 3.3 THE OMPOUND EFFECT OF ATTENTION MECHANISM COMPUTATION

Fig. 3 illustrates the high-level semantic information in the feature map originates from the attention map generated by the deep block. Considering the previously mentioned approach of extracting global category attention as region-guiding information and the structure of the SAM encoder, this paper proposes an adaptive multi-level global attention fusion method. By introducing learnable parameters to dynamically obtain the attention scores at each fusion layer, the weight assigned to the deep block attention is ensured.

In the coarse stage, the input image $X$ is first mapped to a coarse-grained feature space $G_c$ through a patch embedding operation:

$$G_c = \phi(X), \tag{1}$$

where $\phi$ denotes the coarse-grained mapping function, and each patch corresponds to a region of size $patch\_size_{coarse} \times patch\_size_{coarse}$ in the original image. This partition defines the coarse granularity space in accordance with the granular computing-driven framework and serves as the foundation for subsequent importance estimation.

To evaluate the semantic importance of each coarse patch, we employ an adaptive multi-layer global attention fusion strategy. Specifically, given a list of attention maps from selected transformer layers, the attention from the class token to all other patches is extracted for each layer and head. For the $l$-th layer, this produces a per-layer attention vector:

$$S_l = \frac{1}{H} \sum_{h=1}^{H} A_{\text{cls},h}^{(l)}, \; A_{\text{cls},h}^{(l)} = A^{(l)}[:, :, 0, 1 :] \in \mathbb{R}^{B \times H \times 4096}, \tag{2}$$

where $H$ is the number of attention heads, and $A_{\text{cls},h}^{(l)}$ denotes the attention weights from the class token to all non-class patches for head $h$ of layer $l$. Each $S_l$ is then normalized within each sample using min-max normalization to stabilize the attention distribution. The multi-layer attention vectors are then fused using learnable layer-wise weights $\alpha_l$, which are normalized via softmax to ensure differentiability and dynamic contribution:

$$s_{fused} = \sum_{l=1}^{L} \alpha_l \cdot S_l. \tag{3}$$

This fusion emphasizes deeper layers that encode high-level semantic information, consistent with observations that shallow layers tend to produce unstable attention patterns while deeper layers capture more reliable semantic cues. The resulting fused attention map $s_{fused}$ serves as a coarse importance score for all patches in $G_c$.

Finally, the coarse-grained prediction mask $M_c$ is obtained by applying the soft-threshold function defined in the granular computing-driven framework to modulate the features according to the fused attention scores. This differentiable mechanism adaptively highlights high-response regions, providing clear guidance for subsequent fine-grained analysis. The entire process ensures that the coarse stage effectively aggregates semantic information from multiple transformer layers while remaining fully end-to-end trainable and computationally efficient.

### 3.4 THE SECRET TO REDUCING COMPUTATIONAL COMPLEXITY IN ATTENTION MECHANISM

In the fine stage, the high-response regions selected from the coarse stage are processed at a finer granularity to achieve more precise prediction results. By integrating Swin-style window attention with sparse attention mechanisms, this model efficiently models local structures and edge details. The generated mask encodes latent prompts that directly drive the decoder. Let the fine-grained token set be $\{p_i^{fine}\}$, with the corresponding attention defined as

$$a_i = \text{Attention}\left(p_i^{fine}, \{p_j^{fine}\}_{j \in \Omega(M_c)}\right), \tag{4}$$

where $\Omega(M_c)$ denotes the valid finer token set determined by the coarse mask $M_c$. To describe the attention computation more faithfully to the implementation, let $X_Q$ denote the query tokens in a given window, and $X_{KV}$ denote the key/value tokens, which are modulated by the coarse-stage soft mask $M_c$. The projections are computed as

$$Q = X_Q W_Q, \quad K = X_{KV} W_K, \quad V = X_{KV} W_V. \tag{5}$$

The coarse guidance is applied to K and V in a differentiable manner:

$$K_j' = (1 + \alpha \, m_j) \, K_j, \quad V_j' = (1 + \alpha \, m_j) \, V_j, \tag{6}$$

where $m_j \in [0, 1]$ is the token-level soft mask derived from $M_c$, and $\alpha$ is a learnable scaling factor. This operation amplifies the contribution of tokens highlighted by the coarse stage while suppressing low-response tokens. Since $m_j$ is continuous and differentiable, the process allows end-to-end gradient propagation.

Within each window, the unnormalized attention logits incorporate relative position biases $B_{ij}$:

$$\ell_{ij} = \frac{Q_i \cdot K_j'^{\top}}{\sqrt{d}} + B_{ij}. \tag{7}$$

Pairwise-level guidance (implemented as the outer product of token masks or other soft relation matrices) is applied element-wise to modulate the logits:

$$\tilde{\ell}_{ij} = \ell_{ij} \cdot p_{ij}, \qquad p_{ij} \in [0, 1], \tag{8}$$

where $p_{ij}$ represents the pairwise soft weight between token $i$ and $j$. The attention weights are then obtained via softmax:

$$\text{Attn}_{ij} = \text{softmax}_j(\tilde{\ell}_{ij}), \tag{9}$$

and the output for each token is computed as

$$a_i = \sum_{j \in \Omega(M_c)} \text{Attn}_{ij} \, V'_j. \tag{10}$$

To facilitate cross-window information flow, non-shift window attention is applied first, followed by shifted window attention after cyclically rolling the feature map, and finally reversed. Local importance scores are then derived from these fine-grained representations using their channel-wise norms:

$$s_i^{fine} = \|x_i^{fine}\|_2, \tag{11}$$

and normalized to $[0, 1]$ within each sample. Finally, the threshold post-processing is applied to obtain $M_f$.

### 3.5 SEMANTIC HEAD FOR MULTI-CLASS PREDICTION.

While the original SAM decoder outputs only binary masks, GrC-SAM performs semantic segmentation by attaching a lightweight semantic head to the decoder output. The decoder produces a dense feature map $F_{\text{dec}}$ that fuses the latent mask prompt with image features. We transform this feature map into a multi-class prediction through a $1 \times 1$ convolutional classifier, yielding a score map of size $B \times K \times H \times W$, where $K$ denotes the number of semantic classes. This design enables GrC-SAM to predict per-pixel semantic labels without modifying the SAM image encoder or relying on the encoder's class token. The coarse-to-fine mask $M_f$ serves as a latent mask prompt that guides the decoder toward the correct spatial regions, while the semantic head performs the final class discrimination. In this way, prompt-free mask generation and semantic label prediction are integrated into a unified, end-to-end trainable framework.

## 4 EXPERIMENTS

### 4.1 DATASETS AND EVALUATION METRICS

We conduct experiments on five widely used datasets to comprehensively evaluate the proposed framework. For multi-class semantic segmentation, we adopt **PASCAL VOC 2012**[1], and **ADE20K**[2], which cover diverse object and scene categories with varying levels of complexity. To further verify the generalization ability of our method in binary segmentation, we evaluate on **ISIC**[3] for medical image analysis and **Oxford-IIIT Pet**[4] for natural image segmentation with fine-grained boundaries. For performance assessment, we use mean Intersection-over-Union (mIoU) and Pixel Accuracy (PA) on multi-class datasets, as mIoU has become the standard measure of semantic segmentation while PA provides a complementary global perspective. For binary segmentation, we employ Dice coefficient and IoU, where Dice is particularly sensitive to the overlap quality of predicted masks and ground truth, and IoU provides a stricter region-level measure. This combination of datasets and metrics ensures a fair and comprehensive evaluation across both large-scale scene parsing and fine-grained object delineation.

### 4.2 MULTI-CLASS SEMANTIC SEGMENTATION BENCHMARKS

Table 1 demonstrates that GrC-SAM achieves competitive performance on both ADE20K and PASCAL VOC benchmarks. Traditional CNN-based models such as FCN, DeepLabV3, and LRASPP

---

[1]M. Everingham et al., "The PASCAL Visual Object Classes Challenge," IJCV 2010.

[2]B. Zhou et al., "Scene Parsing through ADE20K Dataset," CVPR 2017.

[3]N. Codella et al., "Skin Lesion Analysis Toward Melanoma Detection," arXiv 2018.

[4]O. M. Parkhi et al., "Cats and Dogs," CVPR 2012.

Table 1: **Baseline Comparison Test Results.** semantic segmentation performance comparison on two benchmarks. results are reported in mean intersection-over-union (mIoU) and pixel accuracy (PA).

| Method | ADE20K | | PASCAL VOC | |
|---|---|---|---|---|
| | mIoU ↑ | PA ↑ | mIoU ↑ | PA ↑ |
| FCN Long et al. (2015) | 41.4 | 84.2 | 62.7 | 90.3 |
| DeepLabV3 Chen (2017) | 44.1 | 87.6 | 67.4 | 92.4 |
| LRASPP Howard et al. (2019) | 41.3 | 85.8 | 65.9 | 91.2 |
| MaskFormer Cheng et al. (2021) | 46.7 | 90.3 | 78.6 | 95.8 |
| SegFormer Xie et al. (2021) | 50.3 | 90.4 | 79.2 | 96.1 |
| HQ-SAM Ke et al. (2023) | 51.5 | 91.0 | 79.3 | 96.1 |
| SAM2 Ravi et al. | **51.8** | **91.7** | 78.9 | 96.0 |
| FastSAM Zhao et al. (2023) | 50.1 | 88.9 | 75.3 | 94.9 |
| GrC-SAM (Ours) | 50.7 | 90.1 | **79.5** | **96.3** |

show moderate mIoU and PA, whereas transformer-based approaches like MaskFormer and Seg-Former benefit from enhanced global context modeling.

GrC-SAM attains the highest mIoU on PASCAL VOC and strong results on ADE20K, highlighting the effectiveness of our granular computing-driven coarse-to-fine framework. The coarse stage identifies high-response regions, guiding the fine stage to focus attention selectively on semantically important areas. By modulating K and V with coarse-stage soft masks, low-response tokens are suppressed while informative tokens are amplified, producing fine-grained representations that improve pixel-wise accuracy and boundary delineation. Compared to SAM2 and HQ-SAM, GrC-SAM's explicit coarse-to-fine hierarchy and differentiable thresholding provide more adaptive, data-driven guidance, particularly beneficial for complex multi-class scenarios such as ADE20K.

### 4.3 BINARY SEMANTIC SEGMENTATION

Table 2 demonstrates that our GrC-SAM achieves competitive performance on both ISIC and Oxford-IIIT Pet datasets. On ISIC, U²-Net Qin et al. (2020) achieves slightly higher Dice and PA scores, reflecting its strong capability in segmenting medical skin lesions where foreground shapes are often compact and well-defined. Nevertheless, GrC-SAM attains comparable performance, indicating that the coarse-to-fine, granular computing-driven mechanism effectively captures fine structures without sacrificing overall accuracy.

Table 2: **Baseline Comparison Test Results.** binary semantic segmentation performance comparison on ISIC and Oxford-IIIT Pet datasets. results are reported in Dice and pixel accuracy (PA).

| Method | ISIC | | Oxford-IIIT Pet | |
|---|---|---|---|---|
| | Dice ↑ | PA ↑ | Dice ↑ | PA ↑ |
| U²-Net Qin et al. (2020) | **90.6** | **95.7** | 89.3 | 94.4 |
| SAM | 59.3 | 63.2 | 72.7 | 86.6 |
| GrC-SAM (Ours) | 69.7 | 65.0 | **89.6** | **97.0** |

On Oxford-IIIT Pet, which involves diverse pet categories with varied fur patterns and poses, GrC-SAM outperforms both U²-Net and SAM, achieving the highest Dice and PA. This improvement highlights the advantage of our framework in leveraging coarse-stage guidance to focus attention on relevant regions while refining fine-grained details. The results collectively validate that the granularity-guided coarse-to-fine attention strategy is generally effective for binary segmentation tasks, particularly in scenarios with complex foreground structures.

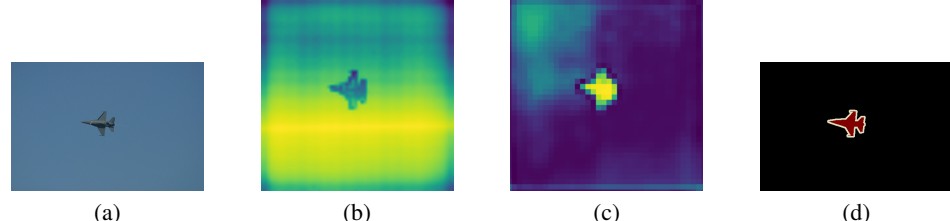

| (a) | (b) | (c) | (d) |

Figure 2: **Granularity visualization.** (a) Input image. (b) Coarse-stage mask capturing the overall aircraft region. (c) Fine-stage mask with clearer structures and boundaries. (d) Ground-truth mask.

## 4.4 ABLATIVE STUDIES

To further demonstrate the effectiveness of our proposed GrC-SAM, we present mask prediction results under coarse prediction and the full refinement process from coarse to fine. Fig. 2b shows the coarse prediction roughly captures the target region but inevitably includes blurred boundaries and background noise. However, Fig. 2d indicates that the fine-grained results significantly improve the delineation of fine details such as the head and legs. After undergoing finer processing guided by the coarse mask, the model generates more precise segmentation with clearer object contours. These comparisons highlight the advantage of introducing a coarse-to-fine granularity process, which enhances mask accuracy and visual quality without requiring external prompts.

To evaluate the effectiveness of our coarse-to-fine design, we compared GrC-SAM with the original SAM using SAM-AMG. Table 3 reports segmentation performance and efficiency metrics on the ADE20K and PASCAL VOC datasets.The results show that GrC-SAM consistently improves multi-class segmentation performance. On VOC2012, mIoU increases by 4.1% and pixel accuracy rises by 2.2%. On ADE20K, GrC-SAM achieves a modest improvement in mIoU while maintaining competitive accuracy.The efficiency gains are particularly notable. GrC-SAM reduces FLOPs by 44% and inference time per image by 87%, demonstrating that the coarse-to-fine framework effectively concentrates computation on high-response regions guided by the coarse stage, avoiding redundant calculations in low-importance areas. Overall, the introduction of coarse-to-fine guidance not only enhances segmentation performance but also significantly reduces computational cost, validating the effectiveness of our hierarchical attention design.

Table 3: **Quantitative Evaluation and efficiency comparison.**

| Method | ADE20K | | VOC2012 | | GFLOPs ↓ | Times ↓ | Params ↓ |
| --- | --- | --- | --- | --- | --- | --- | --- |
| | mIoU ↑ | PA ↑ | mIoU ↑ | PA ↑ | | | |
| SAM (W/O) | 50.3 | 91.2 | 75.4 | 94.1 | 1315.3 G | 1198.87 ms | **93.7** M |
| GrC-SAM (Ours) | **50.7** | **90.1** | **79.5** | **96.3** | **741.5** G | **159.83** ms | 95.7 M |

## 5 CONCLUSION

This paper proposes a coarse-to-fine segmentation framework named GrC-SAM, which integrates granularity computation principles into the foundational SAM model. Through a hybrid hierarchical attention design, our method concentrates computational resources on high-response regions, enabling efficient and precise mask prediction. Extensive experiments on multi-class and binary segmentation benchmarks demonstrate that GrC-SAM consistently outperforms the original SAM model in segmentation quality while significantly reducing computational costs. This research highlights the potential of integrating coarse-to-fine guidance mechanisms and granularity computation into foundational models, paving the way for constructing more efficient and adaptable visual segmentation systems.

ACKNOWLEDGMENTS

We would like to express our gratitude to the large language model (GPT-4) for its invaluable assistance and refinement during the paper writing process.

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

# A APPENDIX

## A.1 PATCH-LEVEL SEGMENTATION AS AN INTERMEDIATE PARADIGM

Existing semantic segmentation methods can be broadly categorized into two paradigms: *mask-level segmentation* and *pixel-level segmentation*. The main distinction lies in the granularity of classification. Mask-level approaches treat each candidate region or proposal as a basic unit, assigning a semantic label to an entire mask $\mathcal{M}$, $f_{\mathrm{mask}} : \mathcal{M} \mapsto y \in \mathcal{C}$, where $\mathcal{M} \subset \Omega$ is a set of pixels within the image domain $\Omega$, and $\mathcal{C}$ denotes the semantic category set. In contrast, pixel-level approaches predict a label for each pixel $p \in \Omega$: $f_{\mathrm{pixel}} : p \mapsto y_p \in \mathcal{C}$.

However, natural images exhibit two structural properties: (1) **semantic sparsity**, as only a small fraction of regions carry discriminative information; and (2) **spatial locality**, as neighboring pixels tend to share similar semantics. Direct pixel-level modeling ignores the spatial redundancy, while mask-level modeling may overlook fine-grained local details. To strike a balance, we propose to segment at the *patch-level*. Specifically, we partition the image into non-overlapping patches $\{P_i\}_{i=1}^N$, where each patch $P_i \subset \Omega$ consists of a group of pixels. The segmentation task is then formulated as $f_{\mathrm{patch}} : P_i \mapsto y_i \in \mathcal{C}$, with the prediction shared across all pixels $p \in P_i$. This formulation can be interpreted as a middle ground between pixel- and mask-level segmentation: $f_{\mathrm{pixel}} \prec f_{\mathrm{patch}} \prec f_{\mathrm{mask}}$, where the notation $a \prec b$ indicates that $b$ captures a coarser granularity than $a$.

From a computational perspective, patch-level segmentation reduces the number of classification units from $|\Omega|$ (all pixels) to $N$ (number of patches), while still retaining sufficient spatial resolution to preserve local details. From a theoretical perspective, if we denote the entropy of semantic labels as $H(\mathcal{C})$, the expected redundancy reduction can be expressed as

$$R = 1 - \frac{H(\{y_i\}_{i=1}^N)}{H(\{y_p\}_{p \in \Omega})}, \tag{12}$$

which quantifies how patch-level grouping leverages spatial correlation to reduce redundant labeling complexity.

**Relation to superpixels.** The idea of grouping pixels into meaningful units resembles the classical notion of *superpixels*, which aggregate pixels with similar low-level properties (e.g., color or texture). However, unlike superpixels that are typically handcrafted and data-independent, our patch-level grouping is learned in a task-driven manner and is integrated into the attention-based mask generator. This makes our patches not only compact structural units but also semantically adaptive.

**Role in our framework.** It is worth noting that in our approach, patch-level representations are not directly used to output the final segmentation maps. Instead, they serve as **mask prompts** that guide the mask decoder towards accurate region delineation. This design choice allows us to benefit from the efficiency and structural alignment of patch-level reasoning, while still leveraging the powerful pixel-level refinement in the final prediction stage. How to directly apply patch-level information to the segmentation process constitutes both a continuation of the work presented in this paper and a direction for our future research.

In summary, by positioning the segmentation unit at the patch-level, we align with the intrinsic semantic sparsity and locality of images, connect naturally with the intuition of superpixels, and enable a principled balance between efficiency and fine-grained accuracy through prompt-based mask generation.

## A.2 TITLE

Figure 3 provides a supplemental analysis supporting our design decision in the main paper. As shown, shallow layers exhibit large fluctuations across samples and fail to form reliable attention patterns, whereas deeper layers demonstrate significantly more consistent and semantically meaningful behavior. This aligns with our empirical finding that deep-layer attention contributes more stable global semantic cues. Therefore, in GrC-SAM, we fuse attention primarily from the deeper encoder layers to obtain a more reliable coarse-level semantic importance map.

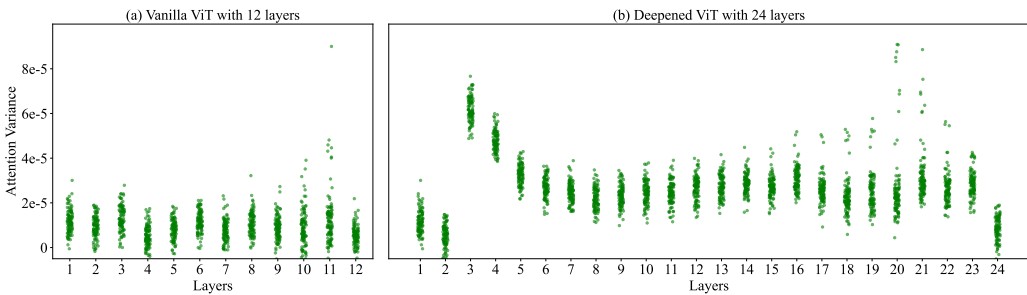

Figure 3: **Attention Variance Display.** Most samples exhibit low variance in the average attention maps across blocks within the standard ViT, indicating that the model has learned stable attention patterns. Some outliers show high variance in deeper layers, suggesting that inter-block information is no longer required at these depths. In deeper ViT architectures, nearly all samples demonstrate significantly higher variance in shallow-layer attention maps, indicating that these layers fail to learn reliable attention patterns Zhang et al. (2023c).

### A.3    MODEL DETAILS

Our Mask Generator is designed to implement a coarse-to-fine segmentation framework, which effectively guides the SAM backbone to focus on high-response regions while preserving fine details. Below we provide a detailed description of the internal feature transformations, patch settings, and the flow of information through the coarse and fine stages.

**Coarse Stage.**    The coarse stage operates on the encoded image features $F \in \mathbb{R}^{B \times C \times H \times W}$. Typically, for input images of size $1024 \times 1024$, after the image encoder, we obtain a feature map of size $F \in \mathbb{R}^{B \times 256 \times 64 \times 64}$. The coarse stage divides this feature map into non-overlapping patches of size $16 \times 16$ pixels in the original image space, resulting in a $64 \times 64$ grid of coarse tokens. A global-guided attention mechanism then uses a fused score map to weigh each patch, generating a soft coarse mask $M_c \in \mathbb{R}^{B \times 1 \times 64 \times 64}$ and updated features $F'_c \in \mathbb{R}^{B \times 256 \times 64 \times 64}$. This mechanism allows the model to allocate attention and computation resources preferentially to semantically significant regions.

**Fine Stage.**    In the fine stage, the coarse feature map $F'_c$ and the soft coarse mask $M_c$ are first upsampled by a factor of 4, yielding finer features $F_{finer} \in \mathbb{R}^{B \times 256 \times 256 \times 256}$ and a sparse guidance mask $M_{finer} \in \mathbb{R}^{B \times 1 \times 256 \times 256}$. Each coarse patch now corresponds to a $4 \times 4$ patch in the finer feature map. A small learnable convolution is applied to $F_{finer}$ to improve interpolation adaptivity. Next, the features are reshaped into tokens of shape $[B, H, W, C]$ and processed by the `RefinedSwinBlock`, which applies local attention within non-overlapping windows, optionally with shift, guided by the sparse mask. The output attention scores are normalized and passed through a learnable threshold selector to produce a soft fined mask, which is finally upsampled to the original resolution $1024 \times 1024$ to generate fined logits for segmentation.

**Patch Settings and Attention Windows.**

- Coarse patches: $16 \times 16$ pixels in image space ($64 \times 64$ coarse grid).
- Fine patches: $4 \times 4$ pixels per coarse patch ($256 \times 256$ fine grid).
- Local attention window size in the fine stage: $6 \times 6$ tokens (Swin-style), sliding to allow cross-window information flow.

**Summary of Feature Shapes.**

- Input image: $B \times 3 \times 1024 \times 1024$
- Encoder output: $B \times 256 \times 64 \times 64$

- Coarse patch tokens: $B \times 256 \times 64 \times 64$

- Soft coarse mask: $B \times 1 \times 64 \times 64$

- Upsampled fine tokens: $B \times 256 \times 256 \times 256$

- Sparse guidance mask: $B \times 1 \times 256 \times 256$

- Final fine logits: $B \times 1 \times 1024 \times 1024$

This hierarchical patch design and attention-guided mechanism ensure that computation is focused on semantically important regions while preserving fine-grained spatial details. By explicitly defining patch sizes and feature transformations at each stage, the Mask Generator efficiently supports our coarse-to-fine framework.

## A.4 TRAIN DETAILS

All images are resized to $1024 \times 1024$ for both training and validation. For training, we apply standard data augmentations including random horizontal flipping (probability $0.5$), random resized cropping with scale range $(0.5, 2.0)$, and color jittering in brightness, contrast, saturation, and hue. Validation only involves resizing and normalization.

The model is trained end-to-end with a composite loss that supervises the coarse, fine, and final predictions. Specifically, the coarse stage is optimized with focal loss to stabilize foreground estimation, the fine stage combines binary cross-entropy and Dice loss to enhance mask quality, and the final stage adopts cross-entropy loss with label smoothing for semantic prediction. The overall objective is a weighted sum of these three components:

$$\mathcal{L} = \lambda_c \, \mathcal{L}_{\text{coarse}} + \lambda_r \, \mathcal{L}_{\text{fine}} + \lambda_f \, \mathcal{L}_{\text{final}},$$

where $(\lambda_c, \lambda_r, \lambda_f) = (0.05, 0.2, 1.0)$.

Optimization is performed using AdamW with an initial learning rate of $1 \times 10^{-4}$, weight decay $1 \times 10^{-4}$, and a cosine annealing schedule. The batch size is set to $4$, and training is conducted for 50 epochs. To stabilize convergence, the image encoder is frozen for the first 5 epochs and then jointly fine-tuned with the rest of the network. All experiments are conducted on a single NVIDIA A100 GPU with 48GB memory.

## A.5 ALGORITHM PSEUDOCODE DETAILS

In the coarse stage, the input image is first encoded into feature maps by the SAM image encoder. These feature maps are divided into coarse patches, and attention maps from selected encoder layers are fused to highlight semantically important regions. A coarse probability map is then generated to indicate the likelihood of foreground regions, serving as a spatial prior for the subsequent fine stage. This stage effectively reduces the search space for refinement by focusing on high-response regions, enabling efficient coarse-to-fine segmentation.

---

**Algorithm 1** The coarse stage with guided attention

---

**Require:** Feature map $F \in \mathbb{R}^{B \times C \times H \times W}$, fused score map $S \in \mathbb{R}^{B \times 1 \times H \times W}$, alpha weights $\alpha$
**Ensure:** Updated feature map $F'$, soft coarse mask $M_c$, coarse threshold $\tau_c$
1: $X \leftarrow \text{Flatten}(F)$
2: $S_f \leftarrow \text{Flatten}(S)$
3: $\alpha_{exp} \leftarrow \text{Interpolate}(\alpha, N)$
4: $\tau_c \leftarrow \text{CoarseThresholdSelector}(S)$
5: $W_{soft} \leftarrow \sigma((S_f - \tau_c) \cdot temp) \odot \alpha_{exp}$
6: $KV \leftarrow X \odot W_{soft}$
7: $X' \leftarrow \text{MHA}(Q = X, K = KV, V = KV)$
8: $F'_{attn} \leftarrow \text{Reshape}(X')$
9: $M_c \leftarrow \text{mean}(W_{soft}, \dim = -1)$
10: $F' \leftarrow \text{ConvFuse}(\text{concat}[F, F'_{attn}])$
11: **return** $F', M_c, \tau_c$

---

In the fine stage, high-response regions identified by the coarse stage are extracted and rescaled to a higher resolution. The extracted patch features are further divided into finer sub-patches, which are processed using window-based sparse attention. Within each window, query vectors are projected from the fine-grained patches, while keys and values are modulated by the coarse-stage soft mask to amplify high-response tokens. The resulting attention outputs are merged to reconstruct fine feature maps, which are then passed through a lightweight feed-forward network with residual connections to produce the final fine segmentation logits. This coarse-guided refinement ensures that fine-grained details are recovered efficiently without processing the entire image at high resolution.

---

**Algorithm 2** The fine stage with local guided attention

---

**Require:** Coarse feature map $F_c \in \mathbb{R}^{B \times C \times H_c \times W_c}$, coarse soft mask $M_c \in \mathbb{R}^{B \times 1 \times H_c \times W_c}$
**Ensure:** Fine logits $F_r \in \mathbb{R}^{B \times 1 \times H_f \times W_f}$, fine threshold $\tau_r$
 1: $F_{up} \leftarrow \text{Upsample}(F_c, \text{scale} = 4)$
 2: $M_{up} \leftarrow \text{Upsample}(M_c, \text{scale} = 4)$
 3: $F_{ref} \leftarrow \text{Conv}(F_{up})$
 4: $tokens \leftarrow \text{Reshape}(F_{ref})$
 5: $sparse\_mask \leftarrow \text{Reshape}(M_{up})$
 6: $X_{attn} \leftarrow \text{RefinedSwinBlock}(tokens, sparse\_mask)$
 7: $A \leftarrow \|X_{attn}\|_2$
 8: $A \leftarrow \text{Normalize}(A)$
 9: $\tau_r \leftarrow \text{fineThresholdSelector}(A)$
10: $M_r \leftarrow \sigma((A - \tau_r) \cdot temp)$
11: $F_r \leftarrow \text{Upsample}(M_r, \text{size} = (H_f, W_f))$
12: **return** $F_r, \tau_r$

---

---

**Algorithm 3** RefinedSwinBlock: sparse Swin-style Attention

---

**Require:** Fine-grained patch features $X \in \mathbb{R}^{B \times C \times H \times W}$, coarse mask $M_c$, window size $ws$, number of heads $\eta$, scaling factor $\alpha$
**Ensure:** fine patch features $X_r$
 1: $X_{win} \leftarrow \text{WindowPartition}(X, ws)$
 2: **for** each window $w$ in $X_{win}$ **do**
 3: $\quad Q \leftarrow \text{Linear}_Q(w)$
 4: $\quad K \leftarrow \text{Linear}_K(w) \odot (1 + \alpha \cdot M_c)$
 5: $\quad V \leftarrow \text{Linear}_V(w) \odot (1 + \alpha \cdot M_c)$
 6: $\quad Q \leftarrow \text{Reshape}(Q, [\eta, M, C/\eta])$
 7: $\quad K \leftarrow \text{Reshape}(K, [\eta, N, C/\eta])$
 8: $\quad V \leftarrow \text{Reshape}(V, [\eta, N, C/\eta])$
 9: $\quad A \leftarrow \text{Softmax}\left(\frac{QK^\top}{\sqrt{C/\eta}} + B\right)$ {Add relative position bias $B$}
10: $\quad w_{out} \leftarrow AV$ {Compute attention output}
11: **end for**
12: $X_r \leftarrow \text{WindowReverse}(w_{out}, ws, H, W)$
13: $X_r \leftarrow \text{MLP}(\text{LayerNorm}(X_r)) + X_r$
14: **return** $X_r$

---

The RefinedSwinBlock implements a window-based sparse Swin attention mechanism on fine-grained patch features. Feature maps are partitioned into non-overlapping windows, and for each window, queries are computed from the window features, while keys and values are modulated by the coarse-stage soft mask with a learnable scaling factor. Multi-head attention is applied within each window with relative position biases to capture local spatial dependencies. Attention outputs are merged via window reversal, followed by layer normalization, a feed-forward network, and residual connection, producing fine patch representations. This design allows the model to selectively focus on salient tokens within each window, guided by coarse-level priors, while keeping computation tractable.

## A.6 Selection of the Fusion Attention Layer

Table 4 and Fig. 4 present the ablation study on feature fusion using four non-consecutive layers selected from a 12-layer encoder. Layers 2, 5, 8, and 11 correspond to global attention layers, while the remaining layers are local attention layers. We observe significant variations in mask prediction across different layer combinations: Configuration A primarily fuses local layers (0, 3, 6, 9), producing minimal background noise but resulting in inaccurate target localization due to insufficient global context. Configuration B (1, 4, 7, 10) distributes selections across local layers yet still introduces noise and fails to provide precise localization. Configuration D (2,5,8,11), integrating all global attention layers, enhances global modeling capability but tends to introduce excessive background regions, increasing mask noise. In contrast, Configuration C (1,4,8,11) achieves the optimal balance between local details and global structure, delivering the most precise segmentation within our coarse-to-fine framework. These results demonstrate that blending local and global attention layers is crucial, and Configuration C's design provides the most effective feature fusion strategy for guiding coarse-to-fine segmentation.

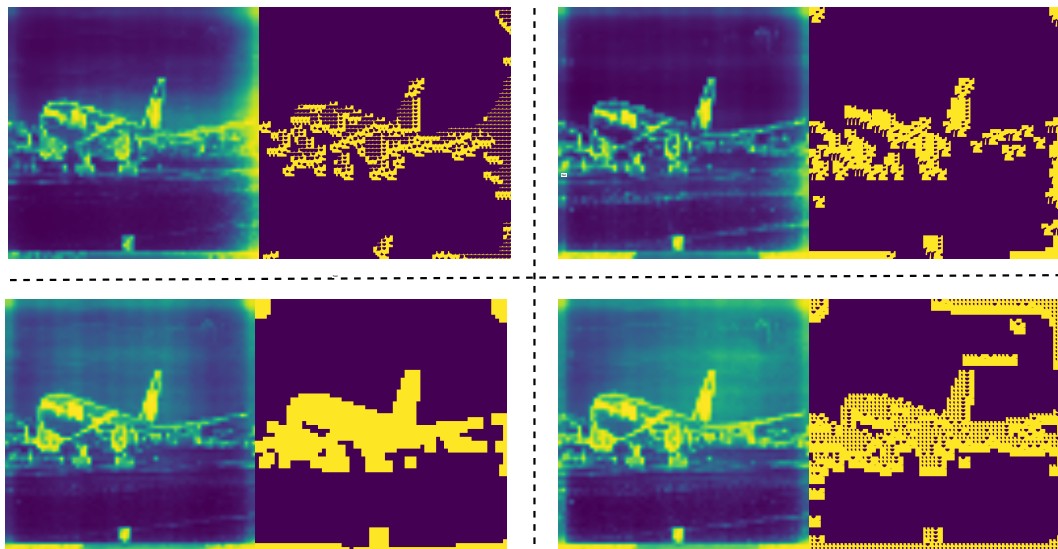

Figure 4: **Attention Fusion Visualization Ablation.** (A) Shallow-only and (B) deep-only fusion each miss either semantic focus or fine structure. (D) Uniform averaging activates background regions. Our learnable multi-layer fusion (C) achieves the best balance between localization and detail.

Table 4: **Ablation study on selecting 4 non-consecutive layers from the 12-layer encoder.**

| Fusion | L0 | L1 | L2 | L3 | L4 | L5 | L6 | L7 | L8 | L9 | L10 | L11 |
|---|---|---|---|---|---|---|---|---|---|---|---|---|
| Config A | ✓ | | | ✓ | | | ✓ | | | ✓ | | |
| Config B | | ✓ | | | ✓ | | | ✓ | | | ✓ | |
| Config C | | ✓ | | | ✓ | | | | ✓ | | | ✓ |
| Config D | | | ✓ | | | ✓ | | | ✓ | | | ✓ |

## A.7 Attention Selection at the Fine-Grained Stage

In the fine-grained stage of attention mechanisms, we compare three different attention mechanisms: Global Attention (MSA), Window Attention (W-MSA), and our proposed Sparse Swin-style Attention (W-SSA). The time complexities of these mechanisms are $O(N^2)$, $O(N)$, and $O(\rho \times N)$, where $N$ is the number of elements in the image, and $\rho$ is the sparsity factor, which represents the proportion of high-response areas we focus on. Specifically, Global Attention (MSA) has a time complexity of $O(N^2)$, where $N = h \times w \times C$ is the combination of image size and the number

Table 5: **Comparison of different attention mechanisms in the fine-grained stage.**

| MSA | W-MSA | W-SSA (Ours) | Time Complexity |
|:---:|:---:|:---:|:---:|
| $\checkmark$ | | | $O(N^2)$ |
| | $\checkmark$ | | $O(N)$ |
| | | $\checkmark$ | $O(\rho \times N)_{\rho<1}$ |

of channels. According to the original paper of the Swin Transformer, the computation formula for global attention is:

$$O(\text{MSA}) = 4hwC^2 + 2(hw)^2C \tag{13}$$

This indicates that global attention requires calculating the relationships between each pixel and all other pixels, leading to a significant increase in computational cost as the image size and the number of channels increase. In contrast, Window Attention (W-MSA) reduces computational costs by dividing the image into non-overlapping small windows. Its time complexity is $O(N)$, where $N = M^2 \times h \times w$, and $M$ is the window size. The computation formula for W-MSA is:

$$O(\text{W-MSA}) = 4hwC^2 + 2M^2hw \tag{14}$$

This computation depends only on the number of elements within the window, significantly reducing the computational cost compared to global attention. Based on this, we propose Sparse Swin-style Attention (W-SSA), which combines the locality of window attention with a sparsification strategy to focus attention computation on high-response areas. The time complexity of Sparse Swin-style Attention is $O(\rho \times N)$, where $\rho$ represents the proportion of the sparse area we focus on. The time complexity derivation process is as follows: we still perform the calculations based on the window attention structure but only operate on the sparse regions. Assuming the image is divided into $M$ windows, and only a portion $\rho$ of each window participates in the calculation, the time complexity of Sparse Swin-style Attention is:

$$O(\text{W-SSA}) = 4hwC^2 + 2\rho M^2hw \tag{15}$$

Here, $\rho$ is the sparsity factor representing the attention to high-response regions, indicating the focus on important semantic information in the image. Table 5 summarizes the time complexities of different attention mechanisms, illustrating the trade-offs between computational efficiency and accuracy.

