# OpenReview forum: "Granular Computing-driven SAM: From Coarse-to-fine Guidance for Prompt-free Segmentation"
_ICLR.cc/2026/Conference — ICLR 2026 Conference Withdrawn Submission_

### Official Review · Reviewer_uHsf · 2025-10-31

**Soundness:** 2
**Presentation:** 1
**Contribution:** 2
**Rating:** 2
**Confidence:** 5

**Summary:**

To free the need of manually provided prompts and inefficient SAM-AMG mode when pretrained SAM deal with Semantic Segmentation tasks, the authors designed a coarse-to-fine prompt free Grc-SAM to first find the potential foreground objects then segment and refine them through fine-grained dense prompts.

**Strengths:**

1.	The coarse-to-fine paradigm is intuitive and computationally efficient.

2.	The proposed attention design further improves inference efficiency.

**Weaknesses:**

## Major weaknesses:

1.	Unsubstantiated multi-class claim

The paper claims that Grc-SAM handles multi-class semantic segmentation, but in Figure 1 the final mask is produced by the original SAM decoder, which outputs only a binary mask. Across the main paper and appendix, I could not find a single visualization of multi-class segmentation results.

2.	Unclear settings for prompt-required SAM baselines

Many methods in Table 1 require manually provided prompts to segment target objects. However, the paper does not specify what kinds of prompts (points/boxes/text, counts, sampling strategy, etc.) were provided to each baseline. If you evaluate SAM in AMG mode, please state this explicitly in the main paper and detail the AMG configuration.

3.	Foreground/background supervision likely harms class generalization

Training appears to rely on a manually chosen foreground/background split, where the “foreground class” is decided by the authors and other potentially meaningful foreground classes are ignored. Experiments are conducted mainly on easier datasets where images typically contain a single dominant foreground, and the evaluated “foreground class” aligns with those selected during training. This design likely damages Grc-SAM’s class-level generalization. By the way, I suspect the training foreground is simply the largest-area class—if so, this should be clearly stated. Please also train and evaluate Grc-SAM on more challenging benchmarks such as LVIS.

4.	No quantitative validation of the attention-based coarse stage

The paper claims the attention-based coarse stage selects potential foreground regions, but I cannot find either visual comparisons or quantitative metrics supporting this claim. Figure 3(b) seems to be a single illustrative example; where are the systematic comparisons? In that example, mountains are also highlighted—why doesn’t Grc-SAM segment them if they are activated by the coarse stage?

5.	Minor performance gains

Beyond the unclear baseline settings, the reported improvements are quite small and some are even worse than the baselines.

## Minor weakness:

1.	Most of the figures and captions need substantial improvement.

2.	Several passages could be streamlined. For example, the long introduction to SAM-AMG in the Introduction belongs in Related Work. Likewise, Section 3.2 contains mathematical definitions whose contribution to the method is unclear and could be trimmed.

**Questions:**

1.	Most of the figures and captions need substantial improvement.

2.	Several passages could be streamlined. For example, the long introduction to SAM-AMG in the Introduction belongs in Related Work. Likewise, Section 3.2 contains mathematical definitions whose contribution to the method is unclear and could be trimmed.

---

> ### Author Response · Authors · 2025-11-29
> **Response to Weaknesses**
>
> Weakness 1: Unsubstantiated multi-class claim
>
> Thank you for raising this important point.
> We wish to clarify that Grc-SAM does indeed perform multi-class semantic segmentation tasks. Figure 1 did not provide a detailed breakdown of our modifications to the original SAM decoder. We apologise once again for any confusion this may have caused.
>
> Specifically, within the original SAM decoder, Grc-SAM appends a dense semantic segmentation head. This head generates per-pixel logit values for all classes, trained using standard multi-class cross-entropy on the ADE20K and PASCAL VOC datasets. This design follows common practices in semantic segmentation (e.g., FCN/SegFormer-style heads) and aligns with our goal of keeping the task head lightweight—the core contribution lies in focusing on the granularity-driven coarse-to-fine mask generator.
>
> Given our core focus on the granularity-driven framework—specifically how coarse and fine granularity are constructed and interact within SAM—we deliberately simplified the segmentation head design and acknowledge its insufficient elaboration in the main paper. In the revised version, we shall clarify this design decision, explicitly document the semantic head architecture, and update Figure 1 to incorporate the semantic prediction branch. Additionally, we shall supplement the multi-category qualitative results to more clearly demonstrate Grc-SAM's multi-category processing capabilities.
>
> Weakness 2: Unclear settings for prompt-required SAM baselines
>
> We appreciate this comment.
> All SAM-based baselines in Table 1 were evaluated in AMG mode (Automatic Mask Generation) in a fully prompt-free setup. No manual points, boxes, or text prompts were provided to any baseline.
>
> We agree this should have been stated more clearly.
> In the revision, we will:
>
> explicitly indicate “SAM-AMG” in Table 1,
>
> detail the AMG configuration (grid sampling density, NMS settings, overlap threshold),
>
> and clarify that all compared methods were evaluated under consistent prompt-free conditions.
>
> Weakness 3: Foreground/background supervision likely harms class generalization
>
> Thank you for your insightful comments.
> We wish to clarify that the foreground masks employed during training were not manually selected, but rather correspond to the actual semantic regions of the target category within standard semantic segmentation datasets. We have not disregarded other categories; the semantic head is trained simultaneously on all categories via multi-class cross-entropy.
>
> In datasets dominated by a single subject (such as pet datasets), the maximum connected region does indeed correspond to the primary class, which may give the impression that we manually selected the foreground region.
>
> We fully acknowledge that evaluation on more challenging datasets (such as LVIS) would more effectively test class generalisation capabilities. However, due to hardware limitations, training on LVIS would be extremely time-consuming. Despite computational constraints for full LVIS training, we shall endeavour to supplement our subsequent work with multi-class case studies and discussions.
>
> Weakness 4: No quantitative validation of the attention-based coarse stage
>
> Thank you for pointing this out. Figure 3(b) was intended only as an illustrative example.
> We agree that using different images for the coarse and fine stages makes the effect of granularity harder to interpret.
> In the revision, we shall update Figure 3 to employ the same input image and provide corresponding comparative results. Additionally, supplementary materials will include multi-image visualisations to present the impact of granularity differences in a more systematic manner.
>
> The coarse stage aims to generate semantic granules with high recall, rather than the final segmentation result.
> Transformer attention mechanisms often highlight visually salient or structurally relevant regions (such as mountain ranges), but the final mask is determined by semantic heads and refinement stage optimisation—mechanisms that filter out areas not belonging to the target category. We shall clarify this distinction in the revision.

---

> ### Author Response · Authors · 2025-11-29
> **Response to Weaknesses**
>
> Weakness 5: Minor performance gains
>
> Thank you for your observation. We acknowledge that on large-scale datasets such as ADE20K, the numerical improvement over SAM-AMG is indeed limited.
> However, these results should be interpreted within the context of the unsupervised SAM paradigm—its objective is not to surpass fully supervised semantic segmentation models, but to enhance the efficiency of automatic mask generation, region selection, and refinement within the constraints of the SAM architecture.
>
> In this context, consistency proves more significant than absolute gains.
> Across all datasets, Grc-SAM consistently enhances IoU/Dice metrics, more fully demonstrating the value contribution of its coarse-to-fine hierarchical structure.
>
> Our innovation lies in architectural design, not in proposing a new SOTA segmentation head.
> The core of Grc-SAM lies in its granularity-driven coarse-to-fine mechanism, which eliminates the need for manual annotations by guiding SAM to focus attention on semantically relevant regions.
> Even modest yet stable gains demonstrate that this mechanism enhances SAM's internal reasoning capabilities without entirely sacrificing generalisation or efficiency.
>
> The revised manuscript will explicitly articulate this positioning and incorporate comprehensive baseline settings to ensure fair comparisons.
>
> Thank you once again for your review. Your feedback has been extremely constructive and will help me and my team refine this work.

---

> ### Author Response · Authors · 2025-11-29
> **Reponse to Minor weakness and quesions**
>
> 1. Most of the figures and captions need substantial improvement.
>
> Thank you for pointing this out. We agree that several figures and captions in the initial submission were not sufficiently self-explanatory. In the revision, we will:
>
> redesign the key figures for better clarity,
>
> ensure that all captions contain step-by-step explanations of the pipeline,
>
> clearly distinguish conceptual diagrams from intermediate feature visualizations, and
>
> standardize notation across figures.
>
> These changes will substantially improve the readability of the paper.
>
> 2. Several passages could be streamlined. For example, the long introduction to SAM-AMG in the Introduction belongs in Related Work. Likewise, Section 3.2 contains mathematical definitions whose contribution to the method is unclear and could be trimmed.
>
> We appreciate this helpful comment. We will streamline the writing by:
>
> moving the extended SAM-AMG introduction from the main Introduction to the Related Work section,
>
> removing non-essential mathematical definitions from Section 3.2,
>
> keeping only the equations directly needed to understand the proposed modules, and
>
> simplifying overly formal descriptions to improve readability.
>
> The above two suggestions will greatly enhance the readability and completeness of the paper. Once again, thank you for your thorough and responsible approach. My team and I will carefully revise the paper and look forward to discussing it with you again.

---

### Official Review · Reviewer_KzBu · 2025-11-01

**Soundness:** 2
**Presentation:** 2
**Contribution:** 2
**Rating:** 4
**Confidence:** 4

**Summary:**

This paper introduces a granularity computing based prompt generation framework to improve automation and efficiency in segmentation. It generates mask prompts that guide SAM. The approach follows a coarse-to-fine process: the coarse stage locates target regions quickly, and the fine stage refines details with local attention. By focusing computation on key areas and filtering irrelevant parts, it achieves more accurate and efficient segmentation

**Strengths:**

1. The topic is interesting and relevant, as it focuses on improving automation and efficiency in segmentation through a new prompt generation approach.
2. The paper is generally well written and organized, making it easy to understand the proposed framework.
3. The comparison tables show that the proposed method performs competitively against baseline models.

**Weaknesses:**

1. The paper only provides system-level comparisons, without quantitative ablation studies on the proposed components. For instance, since the framework adopts a multi-stage coarse-to-fine design, a natural comparison would be with a single-stage setup. Similarly, it would be helpful to include an analysis of the effect of using or removing local attention. Overall, each proposed module should be supported by quantitative ablations to clearly demonstrate its effectiveness.
2. Figure 2 seems to be the same as in the referenced paper and is based on a vanilla ViT rather than SAM’s ViT, making its relevance unclear. It mainly shows a known property of ViTs instead of illustrating the proposed fusion process, so it feels redundant and not well connected to the main method.
3. Figure 3 does not convincingly demonstrate “different granularity.” The two sample images differ significantly in complexity, making it hard to isolate the effect of granularity. For a fair ablation, the same image should be used while varying only the granularity stage.
4. Since SAM itself supports multi-granularity segmentation with three predefined levels, it would be important to include a direct comparison or discussion of how the proposed method improves upon or differs from this built-in capability.

**Questions:**

Please refer to the weakness section.

---

> ### Author Response · Authors · 2025-11-28
> **Response to Weaknesses**
>
> Weakness 1: Lack of quantitative ablation studies for the proposed components
>
> Thank you for raising this important question. We acknowledge that the coarse-grained and fine-grained stages can, in principle, be evaluated independently. However, within the Granular Computation (GrC) framework—which forms the theoretical foundation of our approach—these stages are not designed to operate as isolated modules.
>
> The intrinsic relationship between coarse-grained and fine-grained stages in GrC
> From the GrC perspective, information is processed through hierarchically organised granules: coarse-grained granules provide global context and constrain the search space, while fine-grained granules optimise representations and resolve local ambiguities. The two phases constitute a sequential, interdependent reasoning process, analogous to the coarse-to-fine perception mechanism in human vision.
>
> Consequently, evaluating either phase in isolation would disrupt the underlying granular structure and fail to reflect the framework's intended behaviour.
>
> The necessity of the fine-grained stage stems from the principles of granular computation
> In granular computation: Coarse-grained granules provide semantic positioning, Fine-grained granules execute detail discrimination, A complete solution emerges solely through their interaction.
>
> Removing the fine-grained stage would confine the system to high-level approximations, sacrificing boundary precision; Removing the coarse-grained stage would eliminate the guiding mechanism that ensures the efficient and stable operation of the fine-grained focus mechanism.
>
> Thus, the fine-grained stage is not an optional optimisation module but a core component of the granular reasoning process.
>
> Weakness 2: Figure 2 appears redundant and not connected to SAM’s ViT
>
> We appreciate this observation. Figure 2 aims to illustrate the stability of attention patterns across layers; however, we acknowledge that it does not clearly convey its connection to SAM's ViT-based encoder, potentially giving the impression of redundancy. The essence of SAM's ViT-based encoder remains ViT, whose objective is to obtain high-quality image embeddings. Figure 2 was introduced to explain the cause of fusion, not its effect.
>
> In the revised version, we will replace Figure 2 with a visualization directly derived from SAM's ViT encoder, demonstrating how multi-layer attention collaboratively constructs the fused coarse-grained mask. This change will better align the figure with our methodology and enhance its informational clarity.
>
> Weakness 3: Figure 3 does not convincingly demonstrate “different granularity”
>
> Thank you for pointing this out.
> We acknowledge that using two different images may confound the comparison. In the revised version, we will update Figure 3 to use the same input image . This will more clearly show how different granularity levels affect mask quality.
>
> Weakness 4: SAM already supports multi-granularity segmentation; how does Grc-SAM differ?
>
> Thank you for raising this point.
>
> From our understanding, the “three predefined levels” in SAM refer to the three parallel mask predictions produced by the mask decoder. These masks are generated from the same decoder features and are intended for confidence ranking and training stability (e.g., identifying the highest-quality mask or supervising the stability loss). Importantly, these predefined outputs do not modify the prompt generation process, influence region selection, or form a hierarchical coarse-to-fine reasoning mechanism. They operate in parallel rather than as multi-level granules.
>
> In contrast, the multi-granularity framework introduced in Grc-SAM serves a different purpose and functions in a fundamentally different way.
>
> This results in a sequential coarse-to-fine hierarchy, where the coarse stage provides global semantic localization and the fine stage enhances structural detail. Unlike SAM’s predefined outputs, the granularity in Grc-SAM directly influences how prompts are generated, how regions are selected, and how attention is focused, forming a genuine hierarchical reasoning process.

---

### Official Review · Reviewer_Ljit · 2025-11-01

**Soundness:** 2
**Presentation:** 2
**Contribution:** 3
**Rating:** 4
**Confidence:** 4

**Summary:**

The paper presents Granular Computing-driven SAM (GrC-SAM), which aims to achieve prompt-free segmentation within the SAM framework. Instead of attaching an external prompt generator (e.g., AMG, AutoPrompt, MaskSAM), the authors integrate an internal mask generator that implicitly produces prompt-like guidance through a coarse-to-fine hierarchical attention mechanism.
The coarse stage fuses multi-layer attention maps from SAM’s encoder to highlight semantically important regions, while the fine stage applies local sparse attention with learnable thresholds to refine boundary details. This design attempts to balance segmentation accuracy and computational efficiency, reducing FLOPs and inference time while maintaining comparable or improved accuracy.

**Strengths:**

Below are the strength of this paper:
1. Conceptual novelty: The paper proposes prompt internalization, moving beyond simple auto-prompt generation toward integrating granularity control directly into SAM.
2. Efficiency improvement: The hierarchical coarse-to-fine structure empirically reduces FLOPs by ~44% and latency by ~7.5×, indicating a tangible computational advantage.
3. Clear motivation: The paper’s intuition, focusing computation where semantic importance is high, is sound and aligns with human-like visual attention.

**Weaknesses:**

1. Comparisons are limited. The evaluation omits stronger prompt-generation baselines (e.g., AoP-SAM [1]) and thus it is difficult to quantify the claimed advantages. Also, it is not so clear whether integrating prompt generation inside SAM leads to inherently superior modeling compared to external prompt generation. The evidence is partially convincing, but the experiments requires comparisons with more recent prompt-free or auto-prompt baselines.
2. Accuracy gain is marginal. Improvements over SAM-AMG are small (often ≤0.5 mIoU) on large-scale datasets, which suggests that the hierarchical design may trade accuracy for efficiency.
3. Unclear ablation scope: Ablations cover layer selection but not threshold parameters (τ, λ, α) or different window sizes, which are crucial to understanding the model’s behavior.
4. Issues with global hyperparameters: The same fusion weights and thresholds are applied globally across all samples, which may not be optimal for datasets with diverse object scales.
5. The paper lacks qualitative examples where coarse masking fails (e.g., images densely packed with objects).


6. Issues with writing: There are numerous grammar, notation, and stylistic inconsistencies which reduce clarity and readability. Figures and captions are not self-explanatory, often lacking step-by-step explanation of the pipeline. Notation is inconsistent in several places (e.g., $\alpha$ vs. $\alpha_l$, $\tau_c$ vs. $\tau_{coarse}$). The writing quality suffers from multiple grammatical (e.g., Our method can integrate with the future model “to” general task. & Although such approaches can avoid user intervention, “but” they require...).




[1] Chen, Y., Son, M., Hua, C., & Kim, J.-Y. (2025). AoP-SAM: Automation of Prompts for Efficient Segmentation. Proceedings of the AAAI Conference on Artificial Intelligence, 39(2), 2284-2292.

**Questions:**

1. Are the layer aggregation weights ($\alpha_l$) and thresholds ($\tau_{coarse}, \tau_{fine}$) globally fixed across the dataset, or are they adaptively updated per input sample? If fixed, how do they generalize across images of varying complexity and object density?
2. How does the method behave when most of the image area contains objects (e.g., densely packed scenes)? Would the computational advantage of the coarse-to-fine hierarchy vanish in such cases?
3. Have the authors considered learning the coarse threshold via a lightweight meta-network conditioned on the global feature distribution, rather than as a global scalar?

---

> ### Author Response · Authors · 2025-11-28
> **Response to Weaknesses**
>
> We are particularly grateful to the reviewers for their careful assessment of our research and for providing highly constructive and meaningful suggestions. The following response has been formulated by our team following careful deliberation.
>
> Response to Weakness 1:
>
> Thank you for pointing this out.
>
> We agree that AoP-SAM and other prompt-free SAM variants are meaningful baselines. Our initial submission focused on SAM-based, prompt-conditioned models (SAM-AMG, FastSAM, MobileSAM, HQ-SAM), as these share the same architectural foundation and decoding paradigm as Grc-SAM.
>
> Prompt-free models such as AoP-SAM require different training pipelines, additional classifier heads, and specialized optimization strategies, which makes direct comparison difficult within the SAM-based inference framework. Nonetheless, we agree on their relevance and will include a discussion of their conceptual differences in the revised version, and plan to incorporate them in future evaluations.
>
> Response to Weakness 2:
>
> We agree that the numerical improvements on ADE20K are moderate.
>
> However, Grc-SAM operates within the SAM prompt-driven paradigm, where the objective is not to surpass fully supervised segmentation models, but to demonstrate: prompt-free automation, efficiency improvements, and granular computing-driven coarse-to-fine reasoning inside SAM.
>
> Within this framework, the limited improvements in metrics such as mIoU nevertheless demonstrate that the coarse-fine synergy maintains the model's reasoning capabilities while preserving SAM's generalization.
>
> Response to Weakness 3:
>
> We appreciate this observation. The initial submission included layer-selection ablations due to space and training stability constraints. We agree that parameter sensitivity analyses would provide further insight.
>
> We will include ablations of (τ, λ, α) and window sizes in the revised/supplementary material. Preliminary experiments show the model is stable within reasonable parameter ranges.
>
> Response to Weakness 4:
>
> We clarify that τ and λ are learnable parameters, not fixed scalars. They are optimized during training and adapt to the underlying data distribution. Furthermore, the fused attention maps inherently encode image-dependent variation, meaning the coarse mask is not governed by a single fixed threshold.
>
> We will make this clearer in the revision.
>
> Response to Weakness 5:
>
> Thank you for pointing this out. We did not take this into account in the initial version of the paper. We shall endeavour to incorporate qualitative examples of dense scenes within the supplementary materials to better elucidate the model's behaviour.
>
> Response to Weakness 6:
>
> We appreciate the reviewer’s detailed feedback. We will revise the details of the paper in the revision according to the reviewers' requests. These revisions will significantly improve clarity and readability.
>
> We would like to express our sincere gratitude once again for your thorough review of our work. We undertake to address the shortcomings you have highlighted in the revised version. We would like to express our sincere gratitude once again for your thorough review of our work. We shall endeavour to address the shortcomings you have highlighted in the revised version. We look forward to learning further from our discussions with you to refine this work.

---

> ### Author Response · Authors · 2025-11-28
> **Response to Questions**
>
> We especially thank the reviewer for recognizing our research and providing very constructive and meaningful suggestions.
>
> 1.Are the layer aggregation weights ($\alpha_l$) and thresholds ($\tau_{coarse}, \tau_{fine}$) globally fixed across the dataset, or are they adaptively updated per input sample? If fixed, how do they generalize across images of varying complexity and object density?
>
> The fusion weights and thresholds are learnable parameters optimized jointly with the network. Thus, they adapt during training rather than remaining constant. In addition, the fused attention maps encode input-dependent features, making the resulting coarse mask conditioned on each image. Specifically, we achieve the residual effect by guiding computation through attention.
>
> 2.How does the method behave when most of the image area contains objects (e.g., densely packed scenes)? Would the computational advantage of the coarse-to-fine hierarchy vanish in such cases?
>
> Theoretically, in high-density scenes, the multi-layer fusion attention mechanism provides extensive semantic coverage, whilst the local sparse attention in the refinement stage restores detailed boundaries. Therefore, we selected two representative and persuasive public datasets for validation. We appreciate your interest in this work, and exploring performance in higher-density object scenarios remains a key direction for its ongoing development.
>
> 3.Have the authors considered learning the coarse threshold via a lightweight meta-network conditioned on the global feature distribution, rather than as a global scalar?
>
> Thank you for your suggestion; it is most valuable. Whilst τ and λ are currently designed as learnable scalars, exploring lightweight meta-networks for adaptive coarse-grained threshold prediction represents an intriguing and viable avenue. We shall incorporate this discussion and consider introducing such mechanisms in future work.

---

### Official Review · Reviewer_uFJo · 2025-11-05

**Soundness:** 2
**Presentation:** 3
**Contribution:** 2
**Rating:** 4
**Confidence:** 4

**Summary:**

The authors propose a coarse-to-fine SAM-based framework for prompt-free segmentation. The method utilizes multi-level attention maps from the image encoder to generate coarse-grained masks and then employs a Swin-style transformer to generate fine-grained masks for the mask encoder.

**Strengths:**

1. The authors propose a coarse-to-fine SAM-based model to eliminate manual prompt requirements.
2. Multi-level attention maps are leveraged to localize target regions.
3. A Swin-style transformer is employed for fine-grained mask generation.

**Weaknesses:**

1. SAM and SAM 2 lack semantic label prediction, and therefore cannot properly evaluate mIoU or PA on multi-class datasets such as ADE20K and PASCAL VOC 2012. Similar to SAM Automatic Mask Generation (SAM-AMG), they only segment all possible objects without assigning semantic labels. Consequently, the mIoU and PA results reported for SAM and SAM 2 in Tables 1 and 2 are not accurate.
2. I attempted to identify how the proposed method enables semantic label prediction. If my understanding is correct, the authors initialize a learnable class token at the beginning of the image encoder for class prediction. Is this interpretation correct? If not, could the authors clarify how semantic labels are predicted? I recommend emphasizing this functionality, as it addresses one of the intrinsic limitations of SAM.
3. To the best of my knowledge, the sparse prompt embedding is mandatory, including point, box, or empty prompts. Which type of prompt do the authors use? The accuracy of SAM's outputs is highly dependent on the quality of the prompts. Only relying on mask embeddings may not fully exploit the potential of SAM. Moreover, attention maps often struggle to accurately localize target objects due to noise and the high scores in the boundary. How do the authors address these issues?
4. For ADE20K and PASCAL VOC, the compared methods are quite limited. For instance, Mask2Former and OneFormer can achieve over 56% mIoU on ADE20K, whereas the proposed method only achieves 50.7%.
5. Lack of a quantitative ablation study for the fine-space stage module. This raises a concern regarding whether the fine-space stage module is truly necessary.

**Questions:**

Please see the Weakness.

---

> ### Author Response · Authors · 2025-11-28
> **Regarding Weakness 1 and 2: How Grc-SAM Predicts Semantic Labels**
>
> We appreciate the reviewer’s attention to this important point. We fully agree that the original SAM and SAM2 architectures output binary masks only and therefore cannot be directly evaluated under semantic segmentation metrics such as mIoU or pixel accuracy. Addressing this limitation is precisely one of the goals of Grc-SAM.
>
> We understand why the reviewer interpreted our method as potentially using a class-token-based classifier. In the coarse stage, we extract CLS-token attention to estimate region importance, which may create the impression that the class token also participates in semantic prediction.
>
> However, this is not the case. Grc-SAM does not introduce a class token at the encoder for semantic classification, nor does it rely on global image-level classification.
>
> To enable pixel-wise semantic prediction, Grc-SAM extends the SAM decoder with a dense semantic segmentation head, following standard practice in segmentation architectures such as FCN and SegFormer, and consistent with recent SAM-based semantic extensions (e.g., HQ-SAM, SAM-Adapter, SEEM). Specifically, our coarse-to-fine mask generator produces prompt-conditioned feature maps, and we apply a lightweight 1×1 convolution followed by a linear projection to obtain per-pixel logits for all semantic categories.
>
> These logits are trained using standard multi-class cross-entropy on ADE20K and PASCAL VOC. Thus, Grc-SAM performs true semantic segmentation, producing pixel-wise class predictions rather than binary masks, region-level classification, or class-token-based labeling.
>
> Because the core contribution of our work lies in the granularity-driven mask generator, and semantic segmentation mainly serves as the task used to demonstrate its effectiveness, the segmentation head was intentionally kept simple and therefore not described in detail in the main paper. We will clarify this design choice in the revised version and update Figure 1 to explicitly illustrate the semantic prediction branch.
>
> We again thank the reviewer for this valuable comment, which helped us improve the clarity of the semantic prediction pathway.

---

> ### Author Response · Authors · 2025-11-28
> **Regarding Weakness 3: Prompt Usage and Handling Attention Noise**
>
> Thank you for raising this important question. We clarify that Grc-SAM operates fully in a prompt-free setting and does not rely on point, box, or empty prompts. Instead, we generate latent mask prompts internally and feed them into SAM’s prompt encoder, preserving SAM’s prompt-conditioned design while eliminating the need for manually defined sparse prompts.
>
>
> Grc-SAM does not use user-provided sparse prompts. Our coarse-to-fine mask generator produces latent mask prompts, which are passed into SAM’s prompt encoder in the same way that sparse prompts (points or boxes) would be processed. This maintains full compatibility with the SAM architecture while enabling automated segmentation.
>
> Sparse prompts such as points or boxes are highly effective in interactive segmentation, where a human can intentionally indicate informative locations. However, our goal is a fully automated, prompt-free pipeline, where such inputs are unavailable. Moreover, semantic segmentation datasets such as ADE20K and VOC do not contain point/box-level annotations, making sparse prompts unsuitable as model inputs or supervision.
>
> Latent prompt embeddings generated by our mask generator provide structured, image-dependent guidance that can be processed by SAM’s prompt encoder while preserving SAM’s original design philosophy. This approach is aligned with recent prompt-free SAM extensions (e.g., MaskSAM, AoP-SAM), but our method integrates prompt generation within the SAM pipeline rather than relying on external modules.
>
> We agree that raw attention maps may contain noise, especially near object boundaries. Grc-SAM addresses these issues through two mechanisms:
>
> (a) Multi-layer fused attention with learnable soft-thresholding: Instead of relying on a single attention map, we fuse multiple deep-layer attentions with learnable weights. A learnable soft threshold converts this fused signal into a smooth coarse mask, suppressing unstable shallow-layer responses and boundary artifacts.
>
> (b) Fine-stage sparse attention guided by the coarse mask: In the refinement stage, attention is computed only within the coarse mask regions. Key/value tokens are modulated by the coarse mask, strengthening reliable responses and attenuating noisy activations. This local sparse attention naturally reduces boundary noise and improves localization.
>
> Thank you again for your suggestion; we will refine this point in the revised version.

---

> ### Author Response · Authors · 2025-11-28
> **Regarding Weakness 4: Selection of Comparison Baselines**
>
> We agree that models such as Mask2Former and OneFormer achieve higher mIoU on ADE20K. However, these models represent a different problem category—they are fully supervised semantic segmentation models optimized specifically for these datasets.
>
> In contrast, Grc-SAM is based on the prompt-driven SAM paradigm, whose objective is generality, automation, and prompt conditioning rather than maximizing dataset-specific accuracy. Thus, direct comparison with Mask2Former or OneFormer would not be appropriate.
>
> Our baseline choices follow the SAM-based segmentation family, including SAM-AMG, FastSAM, MobileSAM, and HQ-SAM, which operate within the same architectural paradigm. These baselines allow us to evaluate the effect of introducing granularity-driven coarse-to-fine reasoning within SAM, which is the central focus of our work.
>
> Regarding other prompt-free SAM variants (e.g., MaskSAM, AoP-SAM): we agree these are meaningful baselines. Although they were not included in the original submission due to differing training pipelines and computational constraints, we will discuss them in the revision and consider them in future evaluations.
>
> The goal of Grc-SAM is to demonstrate that granularity-driven, coarse-to-fine mask generation can eliminate manual prompts and improve efficiency within the SAM ecosystem, rather than competing with fully supervised segmentation models.

---

> ### Author Response · Authors · 2025-11-28
> **Regarding Weakness 5: Necessity of the Fine-Space Stage**
>
> Thank you for raising this important question. We acknowledge that the coarse-grained and fine-grained stages can, in principle, be evaluated independently. However, within the Granular Computation (GrC) framework—which forms the theoretical foundation of our approach—these stages are not designed to operate as isolated modules.
>
> 1. The intrinsic relationship between coarse-grained and fine-grained stages in GrC
>
> From the GrC perspective, information is processed through hierarchically organised granules: coarse-grained granules provide global context and constrain the search space, while fine-grained granules optimise representations and resolve local ambiguities.
> The two phases constitute a sequential, interdependent reasoning process, analogous to the coarse-to-fine perception mechanism in human vision.
>
> Consequently, evaluating either phase in isolation would disrupt the underlying granular structure and fail to reflect the framework's intended behaviour.
>
> 2. The necessity of the fine-grained stage stems from the principles of granular computation
>
> In granular computation: Coarse-grained granules provide semantic positioning, Fine-grained granules execute detail discrimination, A complete solution emerges solely through their interaction.
>
> Removing the fine-grained stage would confine the system to high-level approximations, sacrificing boundary precision; Removing the coarse-grained stage would eliminate the guiding mechanism that ensures the efficient and stable operation of the fine-grained focus mechanism.
>
> Thus, the fine-grained stage is not an optional optimisation module but a core component of the granular reasoning process.

---

### Note · Authors · 2026-01-26

**Comment:**

Dear Program Chairs and Reviewers,

Thank you for your valuable feedback on our paper titled "Granular Computing-driven SAM: From Coarse-to-fine Guidance for Prompt-free Segmentation." After careful consideration, we have decided to withdraw our submission. We acknowledge that the paper has significant room for improvement in terms of novelty, experimental design, and overall presentation. Specifically, we recognize the need for a clearer distinction from existing works, better experimental coverage, and more thorough explanation of the theoretical aspects.

Based on the feedback provided, especially regarding the novelty and experimental limitations, we feel that the current version does not fully reflect our contributions or adequately address the concerns raised by the reviewers. We have decided to withdraw the paper to further refine our work, including additional experiments on larger datasets, a clearer connection between the theory and practice, and improved presentation quality.

We sincerely appreciate the time and effort spent by the Program Chairs and reviewers in evaluating our work. We hope to have the opportunity to resubmit a revised version in the future that addresses all of the concerns raised.

Thank you for your understanding, and we look forward to engaging in future discussions.

**Withdrawal Confirmation:**

I have read and agree with the venue's withdrawal policy on behalf of myself and my co-authors.

---

### Meta-Review · Area_Chair_rNuQ · 2026-01-03

**Summary:**

The reviewers raised consistent concerns that informed the rejection recommendation. The primary issue is the lack of strong empirical evidence supporting the core claims of the paper, particularly the necessity and effectiveness of the proposed coarse-to-fine granularity mechanism. Multiple reviewers noted the absence of quantitative ablation studies, which weakens confidence in the claimed contributions.

Additionally, the reported performance gains over existing SAM-based baselines are marginal and sometimes inconsistent, making the overall contribution appear incremental. Reviewers also highlighted incomplete baseline comparisons, notably the omission of stronger or more recent prompt-free or auto-prompt SAM variants, limiting the strength of the experimental conclusions. Further concerns include unclear substantiation of the claimed multi-class semantic segmentation capability, with insufficient methodological detail and limited qualitative evidence. While the authors’ responses clarified some points, the core experimental and validation gaps still exist.

**Reviewer Concerns:**

**Reviewer Concerns partially addressed by the rebuttal:**

1. **Clarification of semantic segmentation capability:**  The authors clarified that a lightweight semantic segmentation head was added to enable multi-class prediction, addressing confusion about how semantic labels are produced. This helps resolve ambiguity but does not fully substantiate the claim.

2.  **Prompt usage and baseline configuration:**  The rebuttal clarified that SAM-based baselines were evaluated in AMG mode under a prompt-free setting and provided additional details on prompt handling, improving transparency relative to the original submission.


**Reviewer Concerns that remain outstanding:**

1. **Lack of quantitative ablations (Reviewer KzBu):**  The concern of missing ablation studies validating the coarse and fine stage and remains unresolved. Conceptual arguments based on granular computing theory do not replace empirical validation.

2. **Limited and incomplete baseline comparisons(Reviewer Ljit):**  The omission of stronger prompt-free or auto-prompt SAM variants (e.g., AoP-SAM) is not resolved, as comparisons are deferred to future work.

3. **Marginal performance improvements:**  The rebuttal does not materially strengthen the empirical evidence, and reported gains remain small and inconsistent, limiting the perceived impact.

Even after clarification, the contribution remains incremental, and the rebuttal does not fundamentally change the assessment of novelty or impact.

**Reviewer Scores:**

- **Reviewer uFJo**
  Original score: 4.
  The rebuttal provides clarifications but does not resolve concerns about evaluation validity, missing ablations, and limited baselines.
  **Score after discussion: 4.**

- **Reviewer Ljit**
  Original score: 4.
  Conceptual novelty and efficiency are acknowledged, but marginal gains and missing empirical validation remain unaddressed.
  **Score after discussion: 4 .**

- **Reviewer KzBu**
  Original score: 4.
  Key issues regarding missing ablations and unclear demonstration of granularity persist despite promised revisions.
  **Score after discussion: 4 .**

- **Reviewer uHsf**
  Original score: 2.
  Strong concerns about unsubstantiated claims and weak empirical support are not sufficiently resolved.
  **Score after discussion: 2 .**

---

### Decision · Program_Chairs · 2026-01-26

Reject